# A Novel *COL4A5* Pathogenic Variant Joins the Dots in a Family with a Synchronous Diagnosis of Alport Syndrome and Polycystic Kidney Disease

**DOI:** 10.3390/genes15050597

**Published:** 2024-05-08

**Authors:** Ludovico Graziani, Chiara Minotti, Miriam Lucia Carriero, Mario Bengala, Silvia Lai, Alessandra Terracciano, Antonio Novelli, Giuseppe Novelli

**Affiliations:** 1Department of Biomedicine and Prevention, University of Rome “Tor Vergata”, 00133 Rome, Italy; chiara.minotti95@gmail.com (C.M.); miriamcarriero@gmail.com (M.L.C.); novelli@med.uniroma2.it (G.N.); 2Medical Genetics Unit, Tor Vergata University Hospital, 00133 Rome, Italy; mario.bengala@ptvonline.it; 3Division of Nephrology, Department of Translational and Precision Medicine, “Sapienza” University, 00133 Rome, Italy; silvia.lai@uniroma1.it; 4Translational Cytogenomics Research Unit, Bambino Gesù Children’s Hospital, IRCCS, 00146 Rome, Italy; alessandra.terracciano@opbg.net (A.T.); antonio.novelli@opbg.net (A.N.)

**Keywords:** polycystic kidney disease, Alport Syndrome, *COL4A5*, phenotypic variability, sensorineural hearing loss

## Abstract

Alport Syndrome (AS) is the most common genetic glomerular disease, and it is caused by *COL4A3*, *COL4A4*, and *COL4A5* pathogenic variants. The classic phenotypic spectrum associated with AS ranges from isolated hematuria to chronic kidney disease (CKD) with extrarenal abnormalities. Atypical presentation of the disorder is possible, and it can mislead the diagnosis. Polycystic kidney disease (PKD), which is most frequently associated with Autosomal Dominant PKD (ADPKD) due to *PKD1* and *PKD2* heterozygous variants, is emerging as a possible clinical manifestation in *COL4A3-A5* patients. We describe a *COL4A5* novel familial frameshift variant (NM_000495.5: c.1095dup p.(Leu366ValfsTer45)), which was associated with AS and PKD in the hemizygous proband, as well as with PKD, IgA glomerulonephritis and focal segmental glomerulosclerosis (FSGS) in the heterozygous mother. Establishing the diagnosis of AS can sometimes be difficult, especially in the context of misleading family history and atypical phenotypic features. This case study supports the emerging genotypic and phenotypic heterogeneity in *COL4A3-A5*-associated disorders, as well as the recently described association between PKD and collagen type IV (Col4) defects. We highlight the importance of the accurate phenotyping of all family members and the relevance of next-generation sequencing in the differential diagnosis of hereditary kidney disease.

## 1. Introduction

Alport Syndrome (AS) is the most frequent hereditary disorder of the basement membrane (BM), and it results from defects of collagen type IV (Col4) [1]. X-linked AS (XLAS) accounts for the majority of cases and it is determined by pathogenic variants in *COL4A5* (Collagen, Type IV, α-5; MIM: 303630) [2]. Milder disease can occur in heterozygous women and hemizygous men with specific amino acid substitutions [3]. Autosomal recessive AS (ARAS) results from biallelic variants in *COL4A3* (Collagen, Type IV, α-3; MIM: 120070) or *COL4A4* (Collagen, Type IV, α-5; MIM: 120131); autosomal dominant inheritance (ADAS) is considered rare and is usually associated with a less severe phenotype [1]. Digenic inheritance has also been described in a minority of patients, with variable phenotypic features [4]. More than 5000 pathogenic *COL4A3-COL4A5* variants have been reported to date, among them major rearrangements, truncating, splicing and missense [3,5,6].

Col4 is the main structural component of the BM, as it has a primary role in supporting cell adhesion, migration, proliferation, and differentiation [7]. Six different Col4 genes (*COL4A1-A6*) encode as many distinct isoforms (a1–a6) as possible, organized in a triple helix structure. Col4 networks containing the a3, a4, and a5 chains are critical to the integrity of the glomerular and tubular membranes of the kidney [8]. Patients with *COL4A3-5* pathogenic variants manifest persistent hematuria and chronic kidney disease (CKD). Most men and about 15% of women with X-linked inheritance, as well as most of those with autosomal recessive disease, progress to renal failure [9]. Col4 is also found in the human testis, inner ear, and eye, thus explaining the characteristic extra-renal manifestations such as sensorineural hearing loss (SNHL) and eye abnormalities (i.e., lenticonus and maculopathy) [1].

An atypical renal presentation of AS is sometimes possible, and it can mislead the diagnosis. Notably, affected individuals who present with proteinuria rather than hematuria are often first diagnosed with steroid-resistant nephrotic syndrome or focal segmental glomerulosclerosis (FSGS). Pathogenic *COL4A3-A5* variants contribute to IgA glomerulonephritis [10,11] and are the most common underlying genetic defect in individuals with familial FSGS [12]. Interestingly, polycystic kidney disease (PKD) has been previously reported in few unrelated patients with *COL4A4* and *COL4A5* variants, occasionally even in the absence of other AS typical features [13,14]. In addition, different recent cohort studies have demonstrated an association between pathogenic variants in the *COL4A3* and *COL4A4* genes and the occurrence of bilateral renal cysts [15,16,17]. On the other hand, up to 8% of patients with suspected Autosomal Dominant PKD (ADPKD; MIM: 173900 and 613095), which is the most frequent cause of renal cystic phenotype, do not receive a molecular diagnosis despite comprehensive screening for *PKD1* (Polycystin 1; MIM: 601313) and *PKD2* (Polycystin 2; MIM: 173910) pathogenic variants [18].

Here, we describe a 24-year-old male patient who presented with early-onset PKD, SNHL, and persistent hematuria in whom next-generation sequencing (NGS) analysis identified a *COL4A5* novel frameshift variant. The finding was consistent with the diagnosis of XLAS in the proband, and inheritance was traced from the proband’s mother who had PKD, IgA glomerulonephritis, and FSGS.

## 2. Case Presentation

The patient (III-1), a 24-year-old man, is the first child of Italian non-consanguineous parents (Figure 1). He was referred to our Genetics Department to evaluate his complex nephrological condition. The patient was born at 39 weeks of gestational age. His mother (II-2) reported that the pregnancy was complicated by preeclampsia (PE), proteinuria, and leg edema, for which she was treated with a low-sodium diet and albumin infusions. His birth weight, length, and occipitofrontal circumference (OFC) were 2280 g (−2.43 SDS), 47.5 cm (−1.29 SDS), and 32 cm (−1.94 SDS), respectively. The Apgar score was 9 at 1 min and 9 at 5 min. The proband presented transient neonatal hypocalcemia (8 mg/dL); abdominal and cranial ultrasounds (US) were normal. Microscopic hematuria with dysmorphic red blood cells (RBCs) was first detected at 20 days and became persistent by 2 years of age. Macroscopic hematuria first manifested during a febrile episode in the first year of life, and it still occurs in the case of upper respiratory tract infections. Bilateral moderate SNHL was diagnosed at the age of 5.

These clinical signs led to a clinical suspicion of AS, which was subsequently confirmed by a kidney biopsy performed at the age of 22. The biopsy revealed numerous foamy macrophages in the interstitium and some in Bowman’s capsule. There was a focal inflammatory cell infiltrate (mononuclear cells) associated with the rare atrophic tubules. Moreover, there were 11 glomeruli (1 sclerotic), and the glomerular basement membranes appeared normal (on light microscopy). Immunostaining for Col4 revealed the absence of a3 and a5 chains. At the age of 12, the abdominal US showed multiple bilateral cortical cysts and the patient was diagnosed with PKD. The last abdominal US (24 years of age) showed mildly enlarged kidneys (13 cm on the left, 12 cm on the right), multiple bilateral cysts of varying sizes, with the largest on the left measuring 6.8 × 6.2 cm, and absence of extra-renal cysts (Figure 2). After the diagnosis of AS and PKD, he started therapy with the angiotensin-converting enzyme (ACE) inhibitor enalapril (5 mg), replaced after about two years with ramipril (5 mg, to 10 mg), in association with the xanthine oxidase inhibitor allopurinol (150 mg). Treatment with the sodium-glucose cotransporter 2 (SGLT2) inhibitor dapagliflozin was performed for about 6 months (at 23 years) and then discontinued due to side effects and the lack of benefits. Ophthalmologically, the patient reported an episode of blepharitis at the age of 18, which resulted in myopia and corneal damage in the left eye and astigmatism in the right eye. A few weeks before the evaluation in our Genetics Department, the patient was admitted to the Emergency Department for acute left back pain and macroscopic hematuria, due to the traumatic rupture of a renal cyst associated with Stage 1 Acute Kidney Injury (AKI) on chronic kidney disease (CKD) Stage G2 A3 [19]. At the time of the last evaluation at the age of 24 years, hematochemical tests of the proband showed uric acid within normal range (5.7 mg/dL, reference range 3.4–7 mg/dL), hyperazotemia (97 mg/dL, reference range 10–50 mg/dL), and hypercreatininemia (1.69 mg/dL, reference range 0.7–1.2 mg/dL). The eGFR (Estimated Glomerular Filtration Rate) was 70.5 mL/min/m^2^, according to the CKD EPI (Chronic Kidney Disease Epidemiology Collaboration) formula [20].

The patient’s mother (II-2), a 51-year-old woman, reported a history of persistent microscopic hematuria with dysmorphic RBCs (first detected as a child), persistent mild proteinuria and macroscopic hematuria during febrile episodes, for which she received a clinical diagnosis of FSGS. Abdominal US, performed at 27 years and repeated at 39 years of age, showed multiple bilateral cortical cysts, consistent with a clinical diagnosis of PKD (Figure 2). Abdominal magnetic resonance imaging (MRI) at 48 years confirmed evidence of multiple bilateral cortical cysts, some spontaneously hyperintense on T1WI, with the largest being 2 cm in the middle third of the left kidney (Bosniak class II) [21]. At 31 years of age, she presented IgA glomerulonephritis (diagnosis confirmed with kidney biopsy).

No other family members received a clinical diagnosis of PKD; however, the family history in the maternal line was positive for SNHL in a male cousin of the proband (III-4), and for microscopic or macroscopic persistent hematuria in numerous female members (Figure 1).

After genetic counseling, informed consent was obtained from the proband and his parents, who underwent diagnostic molecular tests. NGS analysis was performed using the Nephropathy Solution (NES) kit (Sophia Genetics, Lausanne, Switzerland) covering the coding regions of 44 most clinically relevant genes related to a broad range of nephropathies (*AGXT*, *AQP2*, *ATP6V0A4*, *ATP6V1B1*, *AVPR2*, *BSND*, *CASR*, *CEP290*, *CLCN5*, *CLCNKB*, *COL4A3*, *COL4A4*, *COL4A5*, *CRB2*, *CTNS*, *CUBN*, *CYP24A1*, *DSTYK*, *EMP2*, *EYA1*, *FN1*, *FOXC1*, *GRHPR*, *HNF1b*, *KANK2*, *KCNJ1*, *LAMB2*, *NPHS2*, *NR3C2*, *OCRL*, *PAX2*, *PHEX*, *PKD1*, *PKD2*, *PKHD1*, *SIX1*, *SLC12A1*, *SLC12A3*, *SLC34A1*, *SLC4A1*, *SLC4A4*, *TTC21B*, *UMOD*, *WT1*). Dedicated bioinformatic analysis (SOPHiA DDM platform) allows for the detection of SNV and Indels, and guarantees high performance, even in GC-rich regions and pseudogenes. The NGS analysis revealed a NM_000495.5: c.1095dup p.(Leu366ValfsTer45) frameshift variant in *COL4A5*, which was inherited from the mother (Appendix A). This variant was classified as “likely pathogenic” according to the American College of Medical Genetics and Genomics (ACMGs) guidelines [22] and submitted to the ClinVar database (http://www.ncbi.nlm.nih.gov/clinvar/, accessed on 6 May 2024, accession ID: SCV005040957.1). In order to exclude any involvement of other PKD-related genes not included in the targeted panel [23], we performed a whole exome sequencing approach using ClinEx pro Extended Clinical Exome kit (4bases, Manno, Switzerland) on the NovaSeq6000 platform (Illumina, San Diego, CA, USA).

## 3. Discussion

The presented case demonstrates a familial *COL4A5* novel pathogenic variant (NM_000495.5: c.1095dup) in association with AS in the hemizygous proband as well as with IgA glomerulonephritis and FSGS in the heterozygous mother. In addition, both the proband and his mother were diagnosed with PKD by the age of 12 and 27, respectively. To the best of our knowledge, the c.1095dup variant has not been previously reported in the scientific literature or in reference genomic databases (e.g., Clinvar, HGVD, gnomAD). The single nucleotide insertion produces a frameshift variant in the *COL4A5* gene which is predicted to cause nonsense-mediated decay (NMD). The involved exon (19 of 53) affects two functional domains: the UniProt protein CO4A5_HUMAN region of interest ‘Disordered’ and the UniProt protein CO4A5_HUMAN region of interest ‘Triple-helical region’. In addition, loss of function (LOF) is a known pathological mechanism in XLAS, and more than 500 pathogenic LOF variants in *COL4A5* have been reported to date [3,6].

Males with XLAS inevitably progress to ESRD, with the overall risk increasing with age [9]. Although available treatment options may delay the onset of renal impairment by limiting the progression of proteinuria and kidney disease, most affected individuals will ultimately require dialysis or a kidney transplant [24]. Genotype–phenotype correlation has been demonstrated with XLAS; in particular, more severe or early onset renal phenotypes are found in male carriers of *COL4A5* large deletions or truncating variants resulting in lower or absent levels of functional protein, with 90% risk of ESRD by age 30 [25,26]. The effects of the *COL4A5* genotype on age at ESRD are not clear in females with XLAS, likely due to the random process of X inactivation (Lyonization) that explains why heterozygous females typically have a variable and generally less severe expression of X-linked recessive disorder than male relatives [27]. Interestingly, most women with progressive CKD due to XLAS do not have hearing loss but still have a significant probability of progression to end-stage renal disease (ESRD) [9].

Recently, increasing phenotypic heterogeneity has been recognized in association with Col4 defects, in addition to the classic attributable renal and extrarenal manifestations. Heterozygous pathogenic *COL4A3-A5* variants are commonly found in familial IgA glomerulonephritis, as the thinned glomerular BM could allow for the migration of IgA from glomerular capillaries to the mesangium [28], as well as in familial FSGS, likely due to the altered glomerular BM resulting in the loss of overlying podocytes and the subsequent development of secondary glomerular hyperfiltration [10]. Phenotypic imbalance in AS occurs sometimes even among individuals sharing the same pathogenic variant, and this occurrence may be partially explained by genetic modifiers (such as variant-specific allelic effects, oligogenic heredity or trans-allele effects in female heterozygous carriers), concurrent renal disease, chronic inflammation-mediated tissue damage, and fibrosis [29]. Other contributing factors, such as infection, nephrotoxic agents, smoking, hypertension, diabetes, and obesity [3] were excluded upon clinical examination in the subjects of the current study.

The cystic phenotype has been very rarely reported in the literature in association with AS [13,30]. However, different recent cohort studies have shown a previously unnoticed prevalence of PKD in individuals with heterozygous pathogenic variants in the *COL4A3* and *COL4A4* genes [15,16,17]. To date, the explanation for the cystogenesis in carriers of *COL4A3-5* pathogenic variants needs to be clarified. CKD is frequently associated with the development of multiple and bilateral kidney cysts in its advanced stages; the lesions are usually <0.5 cm in diameter but can be as large as 3 cm [31]. This occurrence is known as acquired cystic disease (ACD) of the kidney, and renal cysts may arise primarily from proximal tubular epithelial cell proliferation due to compensatory hypertrophy in the functional nephrons [32,33]. The cystic phenotype in ADAS has been demonstrated to have a higher prevalence in older patients with declining eGFR [15,16,17]. Therefore, it has been speculated that CKD could be the concur for the development of cysts in these patients [15]. However, cystic phenotype can also occur in young AS individuals with preserved renal function, as in the present case. Whereby, it is likely that other factors are involved in the development of the trait. Albeit the α1α1α2 and α3α4α5 isoforms are not evenly distributed in the tubular and glomerular BM [8], it has been hypothesized that a defect involving a single gene encoding Col4 isoforms may result in abnormally reduced collagen content along the BM, and thus, in the formation of cysts of glomerular or tubular origin [30]. Glomerulocystic kidney disease and the cystic dilatation of Bowman’s capsule and tubules are found in experimental animal models that harbor pathogenic variants in genes encoding Col4, supporting the etiopathogenetic link between BM defects and cystogenesis in AS. [34,35]. Noteworthy, PKD is a recognized phenotypic trait of HANAC (hereditary angiopathy, nephropathy, aneurysms, and muscle cramps) syndrome, an autosomal dominant disorder determined by heterozygous pathogenetic variants in the *COL4A1* (Collagen, Type IV, α-1; MIM: 120130) gene [36].

Cystic renal disease is most often ascribed within the heterogeneous group of chronic kidney diseases to ADPKD, which is determined by heterozygous pathogenic variants in *PKD1* and *PKD2* in the majority of patients [37]. Occasional cysts can occur in approximately one third of the unaffected population over 70, whereas the finding is uncommon in the younger age groups [38]. ADPKD generally manifests a late onset and substantial variability in the severity of renal involvement and extra-kidney manifestations [18]. The diagnosis of ADPKD is usually established in an individual with age-specific kidney imaging criteria and an affected first-degree relative [39]. Nonetheless, in the current case, an extensive molecular and bioinformatic NGS analysis of the genes to date known to be associated with PKD allowed us to exclude the involvement of additive pathogenic variants to the NM_000495.5: c.1095dup p.(Leu366ValfsTer45) in *COL4A5*.

As emerges from recent studies, the presence of smaller cysts that do not alter the size of the kidney and the absence of extra-renal lesions at US examination may be suggestive of Col4 defects rather than ADPKD [13]. Similarly to previous evidence, cystic nephromegaly and extrarenal cystic phenotype were absent in the proband’s mother (II-2); however, at the last renal US examination, the index case (III-1) had mildly enlarged kidneys in association with PKD. Therefore, although other inherited renal conditions (e.g., ADPKD) should be excluded in the first instance in the presence of cystic nephromegaly [16], undoubtedly Col4 ought also to be considered. To date, the association between Col4 defects and PKD has been statistically demonstrated only in carriers of variants in the COL4A3 and COL4A4 genes, yet it was observed that a significant percentage of XLAS cases had a radiological cisternal phenotype similar to that of ADAS patients [17]. It may be reasonable, therefore, to assume that the development of kidney cysts could be considered an incompletely penetrant consequence of XLAS as well. The presence of PKD was associated with a worse kidney outcome in ADAS cases [15,16,17]. While this might also be true in carriers of pathogenic variants in the *COL4A5* gene, this was not evidenced in our patients, possibly partly due to young age and the absence of long-term follow-up.

The diagnosis of cystic renal disease is often difficult on the sole basis of clinical features, family history, and even kidney histopathology. Comprehensive genetic testing allows for the accurate and early diagnosis of hereditary cystic kidney disorders. The differential diagnosis of cystic kidney disease at early stages is especially important for the selection of patients who will benefit the most from current therapeutic approaches. While the administration of the selective vasopressin V2 receptor (V2R) antagonist, tolvaptan, is considered the gold standard treatment for ADPKD [40,41], clinical practice recommendations for the treatment of individuals with AS encourage the early use of angiotensin-converting enzyme (ACE) inhibitors that have proven efficient to reduce lifetime risk for ESRD [42,43]. In addition, as discussed above, *COL4A5* genotype data can suggest the renal prognosis and partially predict the timing of ESRD, thus guiding the timing and intensity of intervention [44]. On the other hand, according to current knowledge, kidney cysts in patients with *COL4A3-5* pathogenic variants are generally asymptomatic and do not require specific treatment [16,30]. However, the correct genotypic definition in case of PKD allows for ensuring the correct multisystem follow-up pathway in the patient with AS. In addition, the traumatic rupture of renal cysts, which is a known complication in ADPKD [45], is also possible and should be taken into account in patients with *COL4A* variants, as the present case points out. Finally, genetic testing allows for the identification of asymptomatic female carriers, which is important because of their own and their offspring’s risk of renal failure. All women with XLAS should be offered a life-long targeted screening program and the early introduction of ACE inhibitors [43].

Ultimately, whereas the cystic phenotype is emerging as a finding frequently associated with Col4 defects, the differences between AS and ADPKD are subtle, and sometimes the radiological patterns may be entirely comparable, as in the case presented. Clinical criteria alone may, therefore, be insufficient for the differential diagnosis of phenotypically overlapping PKD syndromes in the context of misleading inheritance and atypical presentation [46]. Persistent microhematuria is not a hallmark of ADPKD and even less so if familial aggregation is present, whereas a personal or family history of hematuria with dysmorphic RBCs, proteinuria, in combination with accurate renal phenotyping, may guide the diagnostic classification toward AS [47].

## 4. Conclusions

In conclusion, the presented case further supports the relevance of NGS in the intricate differential diagnosis of hereditary renal disease and eventually direct tailored therapeutic approaches. It also supports the emerging notion that Col4-associated disorders should be considered when PKD is found. Notably, accurate phenotyping of all family members with a history of either hematuria or glomerulopathy may suggest prioritizing *COL4A3-5* screening in those patients with an early finding of PKD. Further studies are needed to clarify the pathogenesis and the prognostic significance of the cystic phenotype associated with Col4 defects and uncover possible genotype–phenotype correlations.

## Figures and Tables

**Figure 1 genes-15-00597-f001:**
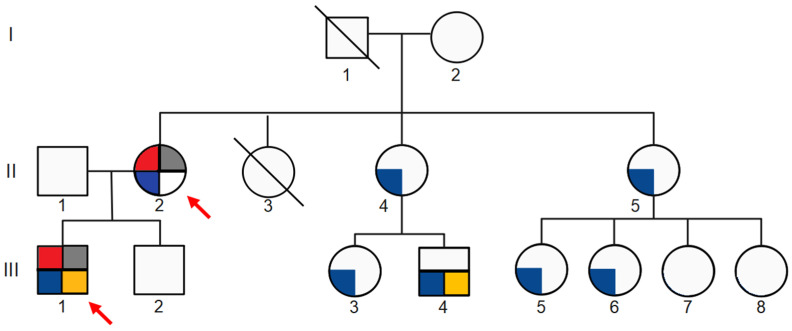
Clinical characterization of the three-generation pedigree of the family. Gray-filled individuals manifested polycystic kidney disease (PKD). Blue-filled individuals manifested either microscopic or macroscopic persistent hematuria. Yellow-filled individuals manifested sensorineural hearing loss (SNHL). Red-filled individuals manifested proteinuria. The proband (III-1) and his mother (II-2) were screened for the NM_000495.5: c.1095dup p.(Leu366ValfsTer45) variant in the *COL4A5* gene (red arrow).

**Figure 2 genes-15-00597-f002:**
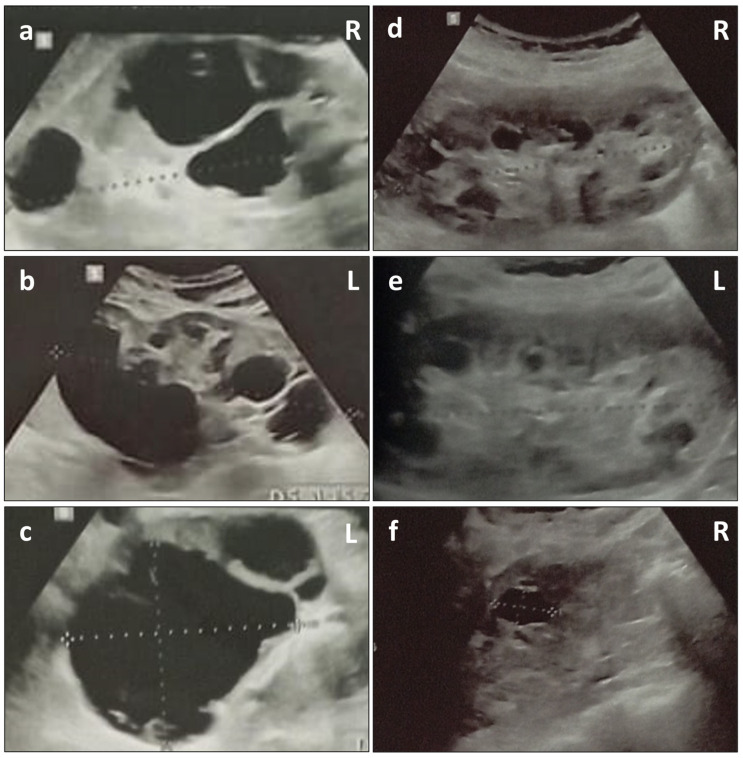
Renal ultrasound (US) of the proband (**a**–**c**) showing bilateral mildly enlarged kidneys (13 cm on the left, 12 cm on the right) and several small-to-large cysts, with the largest on the left measuring 6.8 × 6.2 cm. Renal US of the mother (**d**–**f**) showing kidneys of normal volume (10.7 cm on the left, 10.5 cm on the right) and multiple bilateral cysts of varying sizes, with the largest on the right measuring 1.8 × 1.2 cm. R, right; L, left.

## Data Availability

Data are contained within the article or Appendix A.

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
