# Peer review of "A Novel COL4A5 Pathogenic Variant Joins the Dots in a Family with a Synchronous Diagnosis of Alport Syndrome and Polycystic Kidney Disease"

_genes, 2024, doi:10.3390/genes15050597_

Round 1

Reviewer 1 Report

Comments and Suggestions for Authors

In this manuscript, Graziani et al. describe a 24-year-old man and his mother with X-linked Alport syndrome presenting with multiple bilateral kidney cysts. Genetic testing identified a likely pathogenic variant in COL4A5 gene. The authors claim that this COL4A5 variant is the cause of “a synchronous diagnosis of Alport Syndrome and Polycystic Kidney Disease”.  

Genetic testing was performed by exome sequencing (using Custom Bungle Kit by Sophia Genetics) and in silico analysis of seven genes associated with PKD and AS (COL4A3, COL4A4, COL4A5, PKD1, PKD2, HNF1B, PKHD1).

My major concern is that there is the possibility that patient III-1 has a pathogenic variant in the PKD1 gene that has not been identified. The PKD1 gene is partially duplicated with high sequence similarity at several additional loci on chromosome 16 and standard exome sequencing kits that utilize standard short-read sequence alignment to identify variants have only a 50% sensitivity rate (Ali et al. PMID: 30858458). Authors should explain the coverage and diagnostic rate of PKD1 with the kit they used.

Moreover, other cystic genes could explain the cystic phenotype of patient III-1. There are other genes related to atypical forms of autosomal dominant polycystic kidney disease that should be analyzed: ALG5, ALG9, DNAJB11, GANAB, IFT140, NEK8. For example, individuals with monoallelic loss-of-function variants in IFT140 usually have few large cysts, as patient III-1 seems to have in Figure 2.

On the other hand, the mother's renal ultrasound images are not very clear, in my opinion only a few cysts are distinguishable from the renal pyramids. These few cysts could be normal at 51 years of age.

A more detailed description of the evolution over time of the number of cysts and the kidney volume in the index case is required.

Similarly, a more detailed description of the cystic phenotype would be important in the patient’s mother.

Minor concerns:

-Several recent references regarding kidney cysts in Alport syndrome patients are missing:

Zeni et al. 2024 PMID: 38514012; Bada-Bosch et al. 2024 PMID: 38178635; Furlano et al. 2024 PMID: 38317457.

-In the first paragraph of the discussion, the first 3 sentences of the discussion can be avoided. In the last sentence of this paragraph, it would be better to change cystic renal “disease” by cystic renal “phenotype”. The discussion should be more focus on comparing the case with the recent papers mentioned above.

-As previously mentioned, at least 6 genes in addition to PKD1 and PKD2, have been associated with ADPKD. Consequently, reference 20 might be updated by a more recent one, such as Hanna et al 2023 PMID: 37996359.

-According to the Human Genome Variation Society (HGVS) guidelines, the nomenclature of the variant should be corrected: transcript version number is missing (NM_000495.5); the G after dup should be removed; the protein nomenclature should in brackets. Thus, the nomenclature of the variant should be NM_000495.5: c.1095dup p.(Leu366ValfsTer45) in the whole paper.

-HNF1b should be changed by HNF1B.

Author Response

Please see the attachment for reviewer 1.

Reviewer 2 Report

Comments and Suggestions for Authors

Chronic kidney disease induced by Alport Syndrome has garnered increasing attention in recent years. This condition is driven by mutations in the COL4A3, COL4A4, or COL4A5 genes, imparting a familial hereditary pattern. An understanding of its genetic characteristics significantly aids both prevention and treatment efforts. This paper presents the genotype and phenotype of a patient along with an analysis of their family tree, contributing valuable insights to scientific research. However, there are several areas where the provision of data could be improved to enhance the paper's value significantly:

1. The paper details some clinical information like age, gender, and age at the onset of symptoms thoroughly. However, it lacks adequate clinical kidney information, such as essential renal function data (e.g., eGFR, creatinine, and urea nitrogen). Is there any histological evidence from kidney biopsy or immunohistochemical data on renal damage markers?

2. A family history is provided, and the paper mentions genetic testing performed on the patient's mother. It would be beneficial for the author to include these original gene sequencing results and upload them to a public database such as GEO, facilitating further research by others.

3. Additional pathological examination data would be valuable, particularly concerning any alterations in the basement membrane, to deepen the understanding of the correlation between this mutation and Alport kidney disease.

4. The description of the patient’s treatment is not comprehensive, lacking specifics on treatment measures and medications used.

5. It would be advantageous for the author to provide a prognosis assessment based on the current treatment and progression of the disease. 

These enhancements would not only strengthen the paper but also provide a richer resource for the academic community interested in Alport Syndrome.

Author Response

Please see the attachment (reviewer 2)

Reviewer 3 Report

Comments and Suggestions for Authors

This case report describes a novel frameshift pathogenic variant in the COL4A5 gene (NM_033380.3:c.1095dupG) identified in a 24-year-old male proband presenting with early-onset PKD, sensorineural hearing loss, and persistent hematuria. The variant, inherited from his mother, was associated with Alport Syndrome in the hemizygous proband and with PKD, IgA glomerulonephritis, and FSGS in the heterozygous mother. The family history was positive for SNHL and persistent hematuria in several maternal family members. The case highlights the emerging genotypic and phenotypic heterogeneity in COL4A3-A5-associated disorders, with PKD being a possible clinical manifestation in patients with pathogenic variants in these genes. The authors emphasize the importance of accurate phenotyping of all family members and the relevance of exome sequencing in the differential diagnosis of hereditary kidney diseases, especially when clinical criteria alone may be insufficient due to misleading inheritance patterns and atypical presentations.

Comments:

  1. The study focuses on a single family, which limits the generalizability of the findings.
  2. The study does not provide any functional evidence to support the pathogenicity of the identified COL4A5 variant.
  3. The authors could gather more detailed information about the health status and genetic testing results of other family members, particularly those with SNHL and persistent hematuria.
  4. The case report does not provide information on the long-term clinical outcomes of the proband and his mother.
  5. The study does not compare the clinical and genetic characteristics of this family with other reported cases of PKD associated with COL4A3-A5 variants.  
  6. The authors briefly discuss the potential mechanisms underlying cyst formation in patients with COL4A3-A5 variants but do not provide a comprehensive overview of the current hypotheses.  
  7. The case report does not provide guidance on how the identification of the COL4A5 variant could impact the clinical management of the proband and his family members.
  8. Lack of data on potential genetic modifiers: The authors mention that genetic modifiers could contribute to the phenotypic variability observed in the family but do not provide any data on the presence or absence of such modifiers. To improve, the authors could perform additional genetic analyses to identify potential modifier genes or variants.

Author Response

Please see the attachment (reviewer 3)

Round 2

Reviewer 1 Report

Comments and Suggestions for Authors

The authors have corrected most of my comments.

Reviewer 2 Report

Comments and Suggestions for Authors

The author has supplemented all the data that I thought was missing, so I believe it now meets the publication requirements.